# Oxy210, a Semi-Synthetic Oxysterol, Exerts Anti-Inflammatory Effects in Macrophages via Inhibition of Toll-like Receptor (TLR) 4 and TLR2 Signaling and Modulation of Macrophage Polarization

**DOI:** 10.3390/ijms23105478

**Published:** 2022-05-13

**Authors:** Feng Wang, Frank Stappenbeck, Liu-Ya Tang, Ying E. Zhang, Simon T. Hui, Aldons J. Lusis, Farhad Parhami

**Affiliations:** 1MAX BioPharma Inc., Santa Monica, CA 90404, USA; fwang@maxbiopharma.com (F.W.); fstappenbeck@maxbiopharma.com (F.S.); 2Laboratory of Cellular & Molecular Biology, Center for Cancer Research, National Cancer Institute, National Institutes of Health, Bethesda, MD 20892, USA; liuyatang1@gmail.com (L.-Y.T.); zhangyin@mail.nih.gov (Y.E.Z.); 3Department of Medicine, Division of Cardiology, David Geffen School of Medicine, University of California, Los Angeles, CA 90095, USA; sthui@mednet.ucla.edu (S.T.H.); jlusis@mednet.ucla.edu (A.J.L.)

**Keywords:** Oxy210, oxysterols, inflammation, toll-like receptors (TLRs), anti-inflammatory agents, macrophage polarization, oxysterol therapeutics, white adipose tissue

## Abstract

Inflammatory responses by the innate and adaptive immune systems protect against infections and are essential to health and survival. Many diseases including atherosclerosis, osteoarthritis, rheumatoid arthritis, psoriasis, and obesity involve persistent chronic inflammation. Currently available anti-inflammatory agents, including non-steroidal anti-inflammatory drugs, steroids, and biologics, are often unsafe for chronic use due to adverse effects. The development of effective non-toxic anti-inflammatory agents for chronic use remains an important research arena. We previously reported that oral administration of Oxy210, a semi-synthetic oxysterol, ameliorates non-alcoholic steatohepatitis (NASH) induced by a high-fat diet in APOE*3-Leiden.CETP humanized mouse model of NASH and inhibits expression of hepatic and circulating levels of inflammatory cytokines. Here, we show that Oxy210 also inhibits diet-induced white adipose tissue inflammation in APOE*3-Leiden.CETP mice, evidenced by the inhibition of adipose tissue expression of IL-6, MCP-1, and CD68 macrophage marker. Oxy210 and related analogs exhibit anti-inflammatory effects in macrophages treated with lipopolysaccharide in vitro, mediated through inhibition of toll-like receptor 4 (TLR4), TLR2, and AP-1 signaling, independent of cyclooxygenase enzymes or steroid receptors. The anti-inflammatory effects of Oxy210 are correlated with the inhibition of macrophage polarization. We propose that Oxy210 and its structural analogs may be attractive candidates for future therapeutic development for targeting inflammatory diseases.

## 1. Introduction

Since the discovery of aspirin over 120 years ago, a large array of non-steroidal anti-inflammatory drugs (NSAIDs), steroids, and biologics have been developed and commercialized to halt the detrimental effects of inflammation and the associated pain, however, with chronic use, almost all these agents can cause serious and even life-threatening adverse effects [1,2,3,4]. NSAIDs have been associated with gastrointestinal (GI) damage and increased risk of cardiovascular events. Long-term use of steroids has been shown to contribute to osteoporosis and increased risk of bone fractures, fluid retention in the extremities and the lungs, high blood pressure, psychological effects such as confusion and delirium, thinning of the skin, slowed wound healing, glaucoma, and cataracts [1,4]. Although biologics, such as anti-cytokine therapies and inhibitors of immune cell infiltration into tissues, have added to the arsenal of drugs used to treat inflammation in auto-immune and auto-inflammatory diseases, they may interfere with the ability of the patient to defend against infections and even cancer [1,5]. In some patients, biologics treatment can also result in the formation of antibodies against the therapies (e.g., infliximab), so that response rates may wary and wane over time. The development of novel effective and non-toxic anti-inflammatory drugs, suitable for long-term use, has been challenging but is critically important for the improved management of chronic inflammatory disorders.

In addition to defending against infections, the immune system is also responsible for recognizing and removing endogenous factors that are produced because of inflammation, tissue damage, and aging. In such instances, the first line of defense is the innate immune system controlled largely by tissue macrophages that recognize specific endogenous damage-associated molecular patterns (DAMPs) using pattern recognition receptors (PRRs) that also recognize pathogen-associated molecular patterns (PAMPs) on invading microbes [6,7]. This elicits a response in the host that is mediated by inflammatory cytokines that often result in chronic inflammation and damage to the tissues and organs involved. Perhaps the most recognized and studied PRRs belong to the family of toll-like receptors (TLRs), with TLR4 being the specific receptor for lipopolysaccharide (LPS) that is present on Gram-negative bacteria [8,9]. TLR4 also recognizes endogenous DAMPs such as oxidized low-density lipoprotein (Ox-LDL), molecules damaged by reactive oxygen species, and apoptotic cells, and its engagement results in the production of inflammatory cytokines mediated by co-receptors MD-2 and CD14, MyD88, and adaptors MAL or TRIF [8,9,10,11,12,13]. Given its multitude of roles, the discovery of TLR4 modulators as potential therapies for chronic inflammatory diseases whose pathogenesis appears to involve TLR4, is still an active area despite the inability of TLR4 antagonists such as eritoran to effectively block acute sepsis [14,15,16,17]. Increased TLR expression, activity, and ligands have been demonstrated in several chronic inflammatory diseases such as atherosclerosis, neurodegenerative diseases including Alzheimer’s and multiple sclerosis, dermatologic diseases including psoriasis and bacterial skin infections, chronic neuropathic pain, adipose tissue inflammation associated with obesity, and non-alcoholic steatohepatitis (NASH) to name a few [18,19,20,21,22,23,24,25,26,27,28]. 

Oxysterols are a family of lipid molecules related to cholesterol that can elicit a broad range of biological activities [29,30,31,32,33]. The use of biologically active oxysterol derivatives as starting points in the discovery of new medicines is an approach that we have called Oxysterol Therapeutics^®^. With it, the applied biological context and signaling properties of various naturally occurring oxysterols combined with iterative design and testing cycles of semi-synthetic analogs have led us to identify drug candidates in different therapeutic areas, such as orthopedic medicine, cancer, and fibrotic disease. For example, we have previously focused on oxysterols capable of Hedgehog (Hh) pathway signaling modulation, such as Oxy133 and Oxy186, relevant in regenerative medicine (for example, for bone formation via allosteric activation of the Smoothened (Smo) receptor) [34], and in oncology (through inhibition of aberrant Hh signaling, downstream of the Smo receptor and at the level of Gli), respectively [35]. More recently, we have discovered a semi-synthetic proprietary oxysterol, Oxy210 (Figure 1), which is similar to Oxy186 in potently inhibiting Hh signaling but also is a potent inhibitor of transforming growth factor-beta (TGF-β)-mediated responses in fibroblastic cells and A549 non-small cell lung cancer cells [36], and in human hepatic stellate cells (HSC), with relevance in both fibrosis-driven cancers and chronic fibrotic diseases, such as non-alcoholic steatohepatitis (NASH) [37]. Using human APOE*3-Leiden.CETP transgenic mice that develop all the hallmarks of human NASH when placed on a high-fat Western diet [38], we found that oral administration of Oxy210 over 16 weeks significantly ameliorated NASH symptoms in these mice [37]. The beneficial effects of Oxy210 manifested themselves through inhibition of hepatic fibrosis, hepatic inflammatory and fibrotic gene expression, hepatic lipid deposition, and reduction in hepatic apoptosis and circulating inflammatory cytokines, alanine aminotransferase (ALT), and cholesterol [37]. Given the potent disease-modifying effects of Oxy210 in NASH, in subsequent studies, we asked whether Oxy210 can exert direct anti-inflammatory effects in addition to the observed inhibitory effects on hepatic lipid deposition and oxidation, and cellular apoptosis that may indirectly result in reduced hepatic inflammation. 

In the present report, we evaluated whether Oxy210 has anti-inflammatory effects in macrophages in vitro by examining its effects on inflammatory cytokine expression and M1 and M2 macrophage polarization [39,40,41,42,43,44,45,46,47,48,49]. Building on our previous reports of the inhibitory effects of Oxy210 on hepatic inflammation in NASH, we assessed whether Oxy210 inhibits inflammation in white adipose tissue of APOE*3-Leiden.CETP mice fed a high-fat diet [37]. Our findings demonstrate that Oxy210 appears to have anti-TLR and anti-inflammatory effects that may be relevant to the future therapeutic development of Oxy210 for targeting chronic inflammatory diseases, such as NASH, atherosclerosis, arthritis, inflammation associated with obesity, and psoriasis to name a few. Given its unique ability to inhibit both M1 and M2 macrophage polarization, Oxy210 may serve as a promising drug candidate for diseases and disorders that involve chronic inflammation and fibrosis as seen in NASH, idiopathic pulmonary fibrosis (IPF), cystic fibrosis, systemic sclerosis, polycystic kidney fibrosis, and obesity, in addition to fibrotic solid tumors, for example, pancreatic cancer and hepatocellular carcinoma [39,40,50,51,52,53].

## 2. Results

### 2.1. Oxy210 Inhibits White Adipose Tissue Inflammation

We previously reported that oral administration of Oxy210 in hyperlipidemic APOE*3-Leiden.CETP mice fed a high-fat high cholesterol (HFHC) Western diet inhibited hepatic inflammation and fibrosis in a 16-week study [37]. We now show that as previously reported [38], and similar to models of high-fat diet-induced obesity in mice, feeding a high-fat diet to APOE*3-Leiden.CETP mice induced inflammation in the gonadal white adipose tissue suggested by the increased expression of inflammatory cytokines interleukin-6 (*Il6*) and monocyte chemotactic protein-1 (MCP-1 or *Ccl2*) in animals fed a high-fat diet for 16 weeks compared to those fed a standard chow diet (Figure 2). The expression of macrophage marker CD68 was also induced in white adipose tissue by high-fat diet, suggesting an increased infiltration of macrophages (Figure 2). Oxy210 administered orally for 16 weeks significantly inhibited the expression of white adipose tissue *Il6*, *Ccl2*, and *Cd68* (Figure 2). These findings further support the in vivo efficacy of Oxy210 in inhibiting inflammation.

### 2.2. Oxy210 Inhibits LPS-Induced Expression of Inflammatory Genes in Mouse and Human Macrophages

To examine the potential anti-inflammatory effects of Oxy210, cultured RAW264.7 mouse macrophages were pretreated for 24 h with Oxy210 in DMEM containing 0.1% FBS followed by the addition of LPS. After 24 h, the expression of the following inflammatory genes was assessed: *Il6*, tumor necrosis factor-α (*Tnfα*), inducible nitric oxide synthase (*iNos*), inflammasome (*Nlrp3*), and MCP-1 (*Ccl2*) (Figure 3), with IC_50_s in the low micromolar range 0.99–1.73 μM (Table 1). As expected, LPS-induced gene expression was completely blocked by pretreatment with the TLR4 specific antagonist, CLI-095 (1 μg/mL, data not shown). Oxy210 also inhibited LPS-induced IL-6 and TNF-α mRNA expression in human THP-1 macrophages (Figure 4).

Depending on the structure, chemical properties, and biological context, oxysterols can be activators or inhibitors of cellular signaling. We had previously demonstrated that two related naturally occurring oxysterols, 20(*S*)-hydroxycholesterol (20(*S*)-OHC) and 20α,22(*R*)-dihydroxycholesterol (Oxy16) (Figure 5A) can activate or inhibit Hh signaling, respectively, in mesenchymal cells [54,55,56]. While 20(*S*)-OHC stimulates the pathway via allosteric activation of Smoothened (Smo), Oxy16, which does not bind to Smo, inhibits the pathway downstream of Smo in a Smo-independent manner [34,57]. We have further shown that these opposing biological activities correlate with rigid conformational properties exhibited by 20(*S*)-OHC and Oxy16 which either enable or prevent Smo binding [36]. Smo binders, such as cholesterol and 20(*S*)-OHC, prefer an extended conformation of the sterol side chain, both in the crystalline state and when bound to the sterol binding pocket of the Smo protein (Figure 5A) [58,59,60]. By contrast, Oxy16 departs from the extended orientation to adopt a bent conformation mainly due to intramolecular hydrogen bonding between the neighboring hydroxyl groups at C-20 and C-22 (Figure 5A) [36]. We have applied this analysis to semi-synthetic oxysterol analogs with Hh stimulatory and inhibitory properties. For example, Oxy133, a 20(*S*)-tertiary alcohol, is an Hh pathway stimulator with robust osteoinductive properties and displays an extended conformation in the crystalline state (Figure 5B) and in silico, when docked to the Smo binding pocket (unpublished observations). By contrast, Oxy186 and Oxy210, two semi-synthetic oxysterols with strong Hh pathway inhibitory properties, are 20(*R*)-tertiary alcohols and rigidly held in a bent conformation, the result of steric compression between the angular methyl group at C-18 and the substituents at C-20 [35,36]. Accordingly, unlike Oxy133, oxysterols with Hh pathway inhibitory properties, dock poorly with the Smo binding pocket, predicting little or no Smo binding affinity for Oxy16, Oxy186, and Oxy210 (unpublished observations).

As Hh signaling can potentially affect inflammatory processes in different tissues [61,62,63,64,65,66], we decided to examine the relationship, if any, between the Hh inhibitory properties and the anti-inflammatory effects displayed by Oxy210. For that purpose, we studied two close analogs of Oxy210, Oxy43, and Oxy234 (Figure 5B). Oxy210, Oxy43, and Oxy234 share identical sterol and side-chain portions and differ only in the stereochemistry and substitution at C-20, which critically affects the overall shape of the molecules (Figure 5B). To better understand the structural basis of the biological effects, we examined conformational properties using X-ray crystallographic techniques, as sterol molecules tend to crystallize in their most stable conformation [67]. As shown in Figure 5B, Oxy210 is a 20(*R*)-tertiary alcohol displaying a bent side-chain conformation in its crystal structure. The C-20 epimer of Oxy210, Oxy43, is a 20(*S*)-tertiary alcohol displaying an extended side-chain conformation, not unlike the Hh agonist, Oxy133. Oxy234 is a 20(*R*)-secondary alcohol, expectedly displaying an extended side-chain conformation. Accordingly, both Oxy43 and Oxy234 are devoid of significant Hh pathway inhibitory properties and display weak Hh pathway stimulation instead (unpublished observations) [68]. We found that Oxy43 and Oxy234 both possess significant anti-inflammatory properties, with Oxy43′s effects being similar to and Oxy234′s effects slightly weaker compared to Oxy210 (Figure 6). These findings demonstrate that the anti-inflammatory properties of Oxy210, Oxy43, and Oxy234 do not correlate with their ability to modulate Hh signaling in fibroblastic cells and strongly hint at a separate mechanistic origin of anti-inflammatory effects. We further examined and ruled out Hh pathway inhibition as a mechanism by which Oxy210 exerts its anti-inflammatory effects in macrophages based on the following observations: (1) In our hands, the expression of transcription factors that mediate Hh signaling, Gli1, Gli2, and Gli3, were almost undetectable in macrophages in the absence or presence of LPS, (2) LPS did not induce Gli transcriptional activity in macrophages at baseline or with overexpression of Gli1, and (3) at least six additional oxysterol analogs of Oxy210 with anti-inflammatory activity had no effects on Hh signaling in macrophages, or in C3H10T1/2 cells that respond to Shh (unpublished observations). Moreover, the comparable anti-inflammatory properties of the two epimers, Oxy210 and Oxy43, further suggest that the anti-inflammatory properties do not depend on the conformation of the sterol side chain to any significant extent. 

Furthermore, the crystal structures shown in Figure 5 reveal that the C-20 hydroxyl group, which remains in the same position relative to the sterol core in both Oxy210 and Oxy43, but assumes a different relative position in Oxy234, is unlikely to contribute to the anti-inflammatory properties, suggesting that it also does not determine such properties of the oxysterols in any significant way. Rather, it seems likely that other features of the molecular structures, such as the sterol core, are important in this context. It is conceivable, however, that the pyridyl side chain present in Oxy43, Oxy210, and Oxy234, may convey physicochemical properties that favor anti-inflammatory activities (e.g., lower logP, higher solubility, etc.) regardless of their spatial orientation, at least when compared to other sterols. 

We compared the anti-inflammatory effects of Oxy210 to those of dexamethasone, a steroid, ibuprofen, an NSAID, as well as naturally occurring sterols and oxysterols that were previously reported to have pro-inflammatory (7-ketocholesterol and 7β-hydroxycholesterol) or anti-inflammatory effects, (β-sitosterol, pregnenolone, and 25-hydroxycholesterol) [32,69,70,71,72]. Results showed that at 5 μM, Oxy210 was consistently more effective than 5 μM dexamethasone in inhibiting LPS-induced *Il6* gene expression and as effective in inhibiting *Tnfα* expression in RAW264.7 cells (Figure 7A,B), whereas an equal dose of ibuprofen (5 μM) did not have any inhibitory effects (data not shown). The lack of effect of ibuprofen at this low dose is consistent with reports that much higher doses, often in the range of 120 to 240 μM, are needed for in vitro studies of its biological effects. In THP-1 macrophages, compared to Oxy210, dexamethasone had a similar inhibitory effect on LPS-induced IL-6 expression but a lower inhibitory effect on LPS-induced TNF-α expression. (Figure 7C,D). Consistent with published reports, at 5 μM, 7β-hydroxycholesterol and 7-ketocholesterol further enhanced LPS-induced inflammatory response in RAW264.7 cells assessed by stimulation of *Il6* expression (Figure 8). At 5 μM, β-sitosterol had an inhibitory effect on the LPS-induced expression of *Il6* (Figure 8), but not *Ccl2* expression (Figure 8), and 25-hydroxycholesterol and pregnenolone (both at 5 μM) did not show inhibitory activity (data not shown).

Some oxysterols can activate liver X receptor (LXR) activity in certain cell types [72], and published reports suggest that LXR activation may attenuate inflammation [72,73,74,75]. LXR activation, through ABCA1 induction, has been reported to antagonize TLR2, TLR4, and TLR9 by interfering with membrane lipid organization and disrupting the recruitment of the adaptor protein, Myd88 [75]. LXR can also bind in cis to enhancer regions of inflammatory genes that contain LXR binding sites and reduce chromatin accessibility [72,75]. Since we previously reported that Oxy210 does not induce LXR activity in human HepG2 hepatocytes or mouse NIH3T3E1 fibroblasts [36], in contrast to human A549 non-small cell lung tumor cells (unpublished observations), we examined whether Oxy210 induces LXR activity in RAW264.7 cells, as determined by the induced expression of LXR target genes, *Abca1* and *Abcg1*. We found that Oxy210 did not cause a significant increase in the expression of these LXR target genes in RAW264.7 cells after 24 or 48 h and under our experimental conditions that include 0.1% FBS (Figure 9A,B). However, the LXR ligand, TO901317 (TO), significantly induced *Abca1* and *Abcg1* mRNA expression in RAW264.7 cells under these experimental conditions by 2.3-fold and 1.7-fold, respectively (Figure 9A,B). In addition, TO inhibited LPS-induced *Il6* expression, consistent with previous reports of its anti-inflammatory effects (Figure 9C). Interestingly, Oxy210 did significantly induce ABCA1 and ABCG1 expression in human THP-1 macrophages by 9-fold and 53-fold, respectively, after 48 h under similar experimental conditions to those used with murine RAW264.7 cells (data not shown). Given the absence of correlation between LXR activation and anti-inflammatory effects of Oxy210 in RAW264.7 and THP-1 macrophages, these findings suggest that the anti-inflammatory effects of Oxy210 in macrophages are likely through LXR-independent mechanisms. 

### 2.3. Oxy210 Inhibits the Expression of Inflammatory Genes Induced by Synthetic TLR4 and TLR1/2 Agonists

To further assess the effect of Oxy210 on TLR-stimulated inflammatory gene expression, RAW264.7 cells were treated with TLR4 agonist, synthetic monophosphoryl lipid A (MPLAs), TLR1/2 agonist, CU-T12-9, or TLR3 agonist, polyinosinic-polycytidylic acid (Poly(I:C)), a synthetic analog of double-stranded RNA. Treatment with 1 μg/mL of TLR4 and TLR1/2 agonists stimulated the mRNA expression of *Il6* and *Tnfα* after 24 h (Figure 10), whereas no such induction was found with TLR3 agonist (data not shown). Oxy210 significantly inhibited the expression of the inflammatory cytokines by the TLR agonists (Figure 10). 

To further confirm the inhibitory effect of Oxy210 on TLR signaling, an in vitro activity assay was performed using proprietary TLR4 Leeporter^TM^ Luciferase Reporter-HeLa cell line or TLR2 Leeporter^TM^ Luciferase Reporter-HEK293 cell line (Abeomics, Inc., San Diego, CA, USA). It was found that Oxy210 used at 0.5, 1, 2, 5, and 10 μM inhibited the activities of TLR4 induced by 25 ng/mL LPS, and TLR2 induced by 1 μg/mL Pam3CSK4, in a dose-dependent manner (Figure 11). Higher doses of Oxy210 were not used for consistency with the doses used in our studies using macrophages. We recognize that there might be limitations in the sensitivity of the TLR assays that are run using specialized HeLa cells and HEK293 cells in which a plateau in the inhibitory effect of Oxy210 is reached even by using increasing non-toxic concentrations, for example, in the TLR4 assay. However, these assays clearly confirm our findings from gene expression studies and demonstrate that Oxy210 inhibits TLR4- and TLR2-induced signaling. 

Moreover, the effect of Oxy210 on NFκB and AP-1 activity was assessed using NFκB LeeporterTM Luciferase Reporter-HEK293 and AP-1 LeeporterTM Luciferase Reporter-HEK293 cell lines, respectively (Abeomics), activated by 100 ng/mL of the phorbol ester, phorbol 12-myristate 13-acetate (PMA). Pretreatment with Oxy210 used at 0.5 to 10 μM did not have any inhibitory effect on NFκB activity (data not shown) but significantly inhibited AP-1 activity in a dose-dependent manner (Figure 11). 

### 2.4. Effect of Oxy210 on Cyclooxygenases and Steroid Hormone Receptors

Given the key roles of cyclooxygenase enzymes, COX-1 and COX-2, in mediating the anti-inflammatory effects of NSAIDs, and the roles of the steroid hormone receptors, glucocorticoid (GR) and mineralocorticoid (MR) receptors, in mediating the anti-inflammatory effect of steroidal agents, we examined the potential agonistic effects of Oxy210 on these targets using an in vitro assay system for the assessment of COX-1 and COX-2 activity (BPS Bioscience, San Diego, CA) and a screen of 19 human nuclear hormone receptors in both the agonist and antagonist modes (Eurofins DiscoverX, Fremont, CA). Oxy210, used at concentrations of 0.1, 1, and 10 μM, had no effect on COX-1 or COX-2 activity compared to the inhibitory effects of positive controls, SC-560, a COX-1 inhibitor, and DuP-697, a COX-2 inhibitor (Figure 12). Similarly, Oxy210, used at a concentration of 10 μM, did not significantly activate or inhibit GR or MR activity, whereas positive controls dexamethasone (GR agonist), used at 0.4 and 10 μM, and aldosterone (MR agonist), used at 0.04 and 1 μM, caused full activation of their targets (Table 2). Importantly, Oxy210 did not significantly activate or inhibit any of the other 17 nuclear hormone receptors, including estrogen receptor (ER), progesterone receptor (PR), farnesoid X receptor (FXR), and androgen receptor (AR) (data not shown).

### 2.5. Oxy210 Does Not Interfere with the Anti-Inflammatory Effects of TGF-β in Macrophages

TGF-β plays an important anti-inflammatory role in several settings, including in the blood vessel wall, and interference with TGF-β activity using neutralizing antibodies or genetic manipulation has been shown to increase vascular inflammation in mouse models of atherosclerosis and diabetic kidney disease [76,77,78,79]. Since Oxy210 was previously reported to display significant TGF-β inhibitory properties in fibroblastic cells and in A549 lung tumor cells associated in part with the inhibition of Smad2/3 phosphorylation [36], we examined whether Oxy210 could inhibit the anti-inflammatory effects of TGF-β in macrophages by assessing LPS-induced IL-6, CCL2, and iNOS mRNA expression. Treatment of RAW264.7 cells with TGF-β inhibited LPS-induced *Il6*, *Ccl2*, and *iNOS* expression, an effect that was further enhanced, rather than inhibited, in the presence of Oxy210 (Figure 13). As expected, SB431542, a selective inhibitor of TGF-βRI, ALK4, and ALK7, reversed the inhibitory effect of TGF-β on LPS-induced inflammatory cytokine expression (Figure 13), suggesting that Oxy210 does not interfere with TGF-β receptor signaling in macrophages.

In subsequent studies, we found that the inhibitory effect of Oxy210 on TGF-β induced Smad2/3 phosphorylation was absent or quite modest in RAW264.7 cells compared to its reported robust inhibitory effect in A549 cells [36], leaving most, if not all, of the TGF-β induced Smad2/3 phosphorylation intact (Figure 14A,B). This is likely the reason why Oxy210 does not interfere with the anti-inflammatory effects of TGF-β. As expected, SB431542 completely inhibited TGF-β-induced Smad2/3 phosphorylation (Figure 14A,B). Consistent with this observation, Oxy210 robustly inhibited the hepatic expression of inflammatory cytokines as well as their plasma levels in a humanized mouse model of NASH without displaying any toxic effects that could have been expected if Oxy210 had inhibited systemic anti-inflammatory effects of TGF-β over the 16-week study [37].

### 2.6. Oxy210 Inhibits Both M1 and M2 Polarization of Macrophages

Macrophage polarization has been reported to contribute to the pathogenesis of inflammatory disease, fibrosis, and cancer, with the M1 phenotype known as a prominent pro-inflammatory state of macrophage polarization and the M2 phenotype being the major anti-inflammatory, pro-fibrotic, and tumor-promoting polarization state [39,40,41,42,43,44,45,46,47,48,49]. As shown earlier, treatment of RAW264.7 macrophages with LPS induced an M1 phenotype evidenced by the expression of inflammatory cytokines *Il6*, *Tnfα*, *iNOS*, and *Ccl2*, which was inhibited by Oxy210. We found that treatment of RAW264.7 cells with TGF-β, IL-4+IL-13, or LPS alone stimulated the expression of markers of M2 macrophage phenotype, *Il10* and arginase-1 (*Arg1*) (Figure 15A,B), with synergistic effects when cells were treated with LPS+TGF-β (Figure 15,B) or IL-4+IL-13+TGF-β (Figure 15C,D). Interestingly, similar to its effect on M1 polarization, Oxy210 also inhibited M2 polarization of macrophages and the expression of *Il10* and *Arg-1* (Figure 15A–D). The inhibitory effects of Oxy210 on M2 polarization were confirmed in human THP-1 macrophages in which IL-4+IL-13 induced the expression of the reported M2 markers fibronectin (FN-1) and CD206 [80], and Oxy210 inhibited this induction (Figure 16).

## 3. Discussion

In addition to their well-known capacity for regulating cholesterol homeostasis, more recently, the role of oxysterols in the control of immune responses has been recognized [29,32,33,81]. In the present report, we describe the anti-inflammatory properties of semi-synthetic oxysterol derivative, Oxy210, in vivo in gonadal white adipose tissue of APOE*3-Leiden.CETP mice fed a high-fat diet. We also demonstrate the anti-inflammatory effects of Oxy210 and its related analogs in vitro, in cell cultures of murine and human macrophages that are mediated in part through inhibition of TLR4 and TLR2 signaling, and inhibition of AP-1 activity, independent of LXR signaling. However, based on the present studies, at this time we cannot rule out the possibility that some anti-inflammatory effects of Oxy210 in THP-1 cells may be mediated through LXR signaling. AP-1 together with NFκB are downstream mediators of inflammatory cytokine gene expression induced by TLRs. As the anti-inflammatory effects of Oxy210 in vitro appear to be comparable to those of dexamethasone, a powerful steroidal anti-inflammatory agent, and given the encouraging oral bioavailability, tolerability, and efficacy in limiting inflammation observed in vivo [37], we propose that oxysterol-based TLR modulators, such as Oxy210 and related analogs, may serve as novel drug candidates for the treatment of chronic inflammatory diseases that are commonly mediated in large part by macrophage activation. Although our findings clearly demonstrate that cellular responses to TLR1/2 and TLR4 agonists are inhibited by Oxy210, at this time we do not know whether Oxy210 is a TLR antagonist, or whether Oxy210 inhibits one or more of the intracellular mediators of TLR signaling. 

Oxy210 is part of a group of oxysterols that were previously shown to have potent inhibitory effects on Hh and TGF-β-induced responses in various cell types including fibroblasts, A549 lung tumor cells, and hepatic stellate cells [36,37]. Previous reports have proposed the role of Gli3 in mediating TLR-mediated responses, including in macrophages [82,83,84], and pro-inflammatory effects of Hh signaling have also been noted in specific settings [61,62,63,64,65,66]. However, based on the collective findings described here, it does not appear that inhibition of Hh signaling in macrophages plays a role in the anti-inflammatory effects of Oxy210 in these cells since: (1) LPS did not induce Hh signaling in macrophages that could have been inhibited by Oxy210, (2) there was no detectable baseline Hh signaling in macrophages, and (3) oxysterol analogs of Oxy210 with no inhibitory activity toward the Hh pathway, when tested in fibroblastic cells, retained anti-inflammatory activity in macrophages. Moreover, the anti-inflammatory effects of Oxy210 in macrophages are unrelated to inhibition of TGF-β signaling, which we demonstrate to remain largely intact in macrophages in the presence of Oxy210. This cell type-specific action of Oxy210 could provide a safety feature since a generalized inhibition of TGF-β signaling using neutralizing antibodies to TGF-β or genetic manipulation resulted in inflammatory adverse effects in vivo, in animal models of atherosclerosis and diabetic kidney disease, presumably through interfering with the anti-inflammatory effects of TGF-β [76,85,86].

Dual inhibition of TLRs in macrophages, known to drive inflammatory responses to DAMPs and PAMPs, and TGF-β signaling in fibroblasts and myofibroblasts that drive fibrotic responses, may allow us to target a number of human diseases in which these signaling pathways are upregulated to enhance inflammation associated fibrotic scarring, including NASH, idiopathic pulmonary fibrosis, kidney disorders, cystic fibrosis, and adipose tissue fibrosis in obesity, among others. TLRs have been shown to exacerbate the pro-fibrotic effects of TGF-β, for example, in scleroderma and hepatic fibrosis, and to play a role in the pathogenesis of cystic fibrosis, kidney fibrosis, and lung fibrosis [21,22,28,87,88,89,90,91]. Moreover, and in contrast to its anti-inflammatory effects in other tissues, TGF-β/Smad3 signaling causes adipocyte hypertrophy, hyperplasia, and increased production of inflammatory cytokines, which together with TGF-β contribute to obesity-associated adipose tissue fibrosis [92,93]. Oral administration of Oxy210 in mice results in significant Oxy210 exposure detectable in liver, lung, and brain tissue, and given its vascularity, we expect significant exposure in adipose tissue as well. Hence, Oxy210 may be suitable for targeting fibrotic inflammatory diseases in these organs and tissues as well as others (and unpublished observations) [37]. Based on our findings described in this report as well as previous reports, we propose that further assessment of Oxy210 and its analogs for treating human chronic inflammatory diseases is warranted. 

Oxysterols can be activators or inhibitors of cellular signaling, including Hh and TGF-β signaling. We have previously outlined that conformational properties, in particular, the position of the side chain relative to the sterol core, form the structural basis for Hh stimulatory and Hh inhibitory properties of the oxysterols [36]. For example, 20(*S*)-OHC and Oxy133 are Hh pathway agonists that prefer an extended conformation of their sterol side chain [36]. By contrast, Oxy210 displays Hh and TGF-β pathway inhibitory properties in fibroblasts that correlate with a bent conformation of the sterol side chain [36]. However, Oxy43 and Oxy234, which feature the same sterol side chain as Oxy210, are devoid of Hh and TGF-β pathway inhibitory properties, and consistent with their preference for extended conformations, weakly stimulate Hh signaling in fibroblasts, instead (data not shown). As Oxy43, Oxy210, and Oxy234 all exhibit comparable anti-inflammatory activity in macrophages, our studies clearly demonstrate that the conformation of the sterol side chain cannot be a critical factor with respect to their anti-inflammatory properties, even though the exact molecular target(s) remains to be identified, especially given the fact that a number of receptors, co-receptors, and adaptors are involved in mediating TLR signaling. This suggests a separate mechanistic origin of the anti-inflammatory activity and the Hh and TGF-β signaling inhibitory properties in Oxy210 and may hint at the involvement of other structural elements, such as the tetracyclic sterol core, in anti-inflammatory activities, which is supported by the fact that other sterols are known to possess anti-inflammatory properties, including β-sitosterol although at much higher concentrations than Oxy210.

Our reported studies here demonstrate that Oxy210 is a potent inhibitor of macrophage activation in response to LPS, a classic stimulator of macrophage M1 polarization that is associated with an inflammatory phenotype, and in response to IL-4, IL-13, and TGF-β that are classic stimulators of macrophage M2 polarization that is associated with an anti-inflammatory, pro-fibrotic, and tumor-promoting phenotype [39,40,41,42,43,44,45,46,47,48,49]. To our knowledge, dual inhibition of M1 and M2 macrophage polarization by Oxy210 is unique since other compounds in various therapeutic classes, including anti-inflammatory, anti-fibrotic and antimicrobial agents, have been reported to modulate either the M1 or the M2 phenotype of macrophages, but not both. For example, glucocorticoids and aspirin inhibit macrophage differentiation toward a pro-inflammatory, M1, phenotype in several in vitro and in vivo experimental systems, while promoting the M2 phenotype [94,95]. Anti-fibrotic drugs such as pirfenidone and nintedanib inhibit the pro-fibrotic M2 phenotype in experimental models of lung fibrosis [41,42,96], and specific MEK-inhibitor and HDAC-inhibitor drugs were reported to inhibit M2 polarization in an experimental model of wet macular degeneration [97]. Doxycycline, an antibiotic, inhibited M2 polarization and proangiogenic activity of macrophages in vitro and in vivo [98], whereas a combination of antibiotics isoniazid, rifampicin, pyrazinamide, and ethambutol induced M2 polarization and inhibited M1 inflammatory cytokine production in pleural macrophages from patients with tuberculous pleuritis [99]. Finally, β-sitosterol was found to inhibit M1 pro-inflammatory and promote M2 anti-inflammatory polarization of macrophages in an experimental mouse model of rheumatoid arthritis [100]. 

Chronic inflammation is a known driver of pathology, tissue fibrosis, and organ failure, in numerous human diseases, including NASH, IPF, cystic fibrosis, systemic sclerosis, polycystic, and diabetic kidney diseases [39,40,51,52]. In addition, chronic inflammation plays a notable role in cancer progression, for example, in pancreatic and hepatocellular carcinomas and other solid tumors associated with significant intratumoral and peritumoral inflammation [53]. Oxy210, as a dual inhibitor of inflammation and fibrosis, may bring to bear multi-pronged disease-modifying effects in excess of what is achievable with other therapies, through inhibition of inflammatory and pro-fibrotic pathways in macrophages, fibroblasts, and tumor cells. This notion is supported by the ability of Oxy210 to inhibit M1 polarization of macrophages resulting in anti-inflammatory effects, inhibit M2 polarization of macrophages resulting in anti-fibrotic and anti-tumorigenic effects, while also targeting inflammatory and fibrotic events associated with the activation of fibroblasts and tumor cells.

## 4. Materials and Methods

### 4.1. Cell Culture and Reagents

RAW264.7 mouse macrophages and THP-1 human monocytes were purchased from American Tissue Type Culture Collection (ATCC, Rockville, MD, USA) and cultured in DMEM or RPMI containing 10% fetal bovine serum (FBS) and antibiotics, respectively, as previously reported. THP-1 culture medium also contained 0.05 mM β-mercaptoethanol. THP-1 differentiation into macrophages was achieved by treating the cells in their culture medium with 40 nM phorbol 12-myristate 13-acetate (PMA) for 48 h, detached, PMA removed, and cells plated in 12 well plates for experiments. Differentiation into macrophages was confirmed by CD14 mRNA expression. For experiments, when cells were at approximately 80% confluence, FBS concentration was reduced to 0.1% at the time treatments were initiated and for the 48 h duration of the studies. LPS and PMA were purchased from Sigma-Aldrich (St. Louis, MO, USA), TGF-β from R&D Systems (Minneapolis, MN), MPLAs, CLI-095, and Poly(I:C) from InvivoGen (San Diego, CA, USA), and CU-T12-9, HPI-1, HPI-4, and TO901317 from Cayman Chemical (Ann Arbor, MI, USA). All oxysterols and sterols were purchased from Sigma. Oxysterols were prepared in-house according to our published protocols and those outlined in the present manuscript [34,36].

### 4.2. Synthesis and Molecular Characterization of Semi-Synthetic Oxysterol Derivatives

Materials were obtained from commercial suppliers and were used without further purification. Air or moisture-sensitive reactions were conducted under an argon atmosphere using oven-dried glassware and standard syringe/septa techniques. The reactions were monitored on silica gel TLC plates under UV light (254 nm) followed by visualization with Hanessian’s staining solution. Chromatographic purifications were performed using a Teledyne ISCO automated chromatography system. NMR spectra were measured in CDCl_3_. The data are reported as follows in ppm from an internal standard (TMS, 0.0 ppm): chemical shift (multiplicity, integration, coupling constant in Hz).

#### 4.2.1. Oxy43

Oxy43 was prepared in three synthetic steps as depicted in Figure 17. Briefly, pregnenolone was acylated with benzoyl chloride and the resulting product reacted with the lithium salt of 3-ethynylpyridine. The resulting propargyl alcohol was hydrogenated to the 20(*S*)-tertiary alcohol, Oxy43, using Lindlar’s catalyst. The crude product was purified by chromatography on silica. ^1^H NMR (400 MHz, CDCl_3_) δ 8.35 (2H, broad m), 7.67 (1H, d, J = 6.4 Hz), 7.33 (1H, broad m), 5.33–5.32 (lH, m), 3.35 (1H, dddd, J = 11.0, 11.0, 4.8, 4.8 Hz), 2.65–2.63 (2H, m), 2.29–1.49 (20H, m), 1.35 (3H, s), 1.25–1.04 (4H, m), 0.99 (3H, s), 0.86 (3H, s). ^13^C NMR (100 MHz, CDCl_3_) δ: 149.5, 146.9, 140.9, 138.2, 136.1, 123.1, 121.4, 74.9, 71.6, 58.3, 56.9, 50.0, 45.1, 42.8, 42.3, 40.2, 37.3, 36.5, 31.8, 31.6, 31.3, 27.7, 26.1, 23.8, 22.5, 20.9, 19.4, 13.7. A 5 mg portion of Oxy43 was dissolved in isopropanol (0.5 mL) and crystallization was induced by slow evaporation of the solvent. Single crystal X-ray diffraction data were collected at 100K on a diffractometer with a Bruker Apex-II CCD detector and a Cu-micro focus source. Crystal data: Monoclinic, a = 5.7874(2) Å, b = 12.7545(4) Å, c = 31.6582(9) Å, α = 90° β = 92.236(2) °, γ = 90°, Vol. = 2335.08(13) Å^3^, Space group = P 21. The final anisotropic full matrix least-squares refinement on F2 converged at R1 = 0.0832, wR2 = 0.0705, GOF = 0.992.

#### 4.2.2. Oxy133

Oxy133 was prepared as previously described [34]. A 5 mg portion of Oxy133 was dissolved in acetone and water (3:1, 0.5 mL) and crystallization was induced by slow evaporation of the solvents. Single crystal X-ray diffraction data were collected at 100K on a diffractometer with a Bruker Apex-II CCD detector and a Cu-micro focus source. Crystal data: Triclinic, a = 7.187(4) Å, b = 12.940(7) Å, c = 15.078(8) Å, α = 69.728(6) ° β = 85.931(7) °, γ = 87.700(7) °, Vol. = 1311.9(12) Å^3^, Space group = P 1. The final anisotropic full matrix least-squares refinement on F2 converged at R1 = 0.0603, wR2 = 0.0471, GOF = 1.036.

#### 4.2.3. Oxy210

The synthesis and molecular characterization of Oxy210, including single crystal X-ray diffraction data, were previously described [36].

#### 4.2.4. Oxy234

Oxy234 was prepared in three synthetic steps as depicted in Figure 18. Briefly, pregnenolone was condensed with nicotinaldehyde to the enone which was reduced to the ketone via hydrogenation using Lindlar’s catalyst. The ketone was reacted with sodium borohydride to afford the 20(*R*)-secondary alcohol, Oxy234. The crude product was purified by chromatography on silica. ^1^H NMR (400 MHz, CDCl_3_) δ 8.47 (1H, d, J = 1Hz), 8.45 (1H, dd, J = 5, 2 Hz), 7.53–7.48 (1H, m), 7.23–7.18 (1H, m), 5.35–5.31 (1H, m), 3.62–3.47 (2H, m), 2.97–2.58 (2H, m), 2.31–2.15 (2H, m), 2.12–1.92 (2H, m), 1.90–1.72 (7 H, m), 1.69–1.36 (6H, m), 133–0.89 (6H, m), 1.01 (3H, s), 0.76 (3H, m); ^13^C NMR (100 MHz, CDCl_3_) δ 149.9, 147.21, 140.9, 137.8, 136.0, 123.4, 121.5, 73.6, 71.7, 56.7, 56.2, 50.1, 42.4, 42.3, 40.1, 38.8, 37.3, 36.5, 31.9, 31.7, 31.6, 28.9, 25.4, 24.5, 21.0, 19.4, 12.4. A 5 mg portion of Oxy234 was dissolved in a mixture of isopropanol and water (2:1, 0.5 mL) and crystallization was induced by slow evaporation of the solvents. Single crystal X-ray diffraction data were collected at 100K on a diffractometer with a Bruker Apex-II CCD detector and a Cu-micro focus source. Crystal data: Monoclinic, a = 12.7975(3) Å, b = 5.99100(10) Å, c = 17.0435(4) Å, α = 90°, β = 100.5649(12) °, γ = 90°, Vol. = 1284.57(5) Å^3^, Space group = P 21. The final anisotropic full matrix least-squares refinement on F2 converged at R1 = 0.0382, wR2 = 0.0416, GOF = 1.071.

### 4.3. Quantitative RT-PCR

Total RNA was extracted with the RNeasy Plus Mini Kit from Qiagen (Hilden, Germany) according to the manufacturer’s instructions. One microgram of RNA was reverse-transcribed using an iScript Reverse Transcription Supermix from Bio-Rad (Hercules, CA, USA) to make single-stranded cDNA. The cDNAs were then mixed with Qi SYBR Green Supermix (Bio-Rad) for quantitative RT-PCR assay using a Bio-Rad I-cycler IQ quantitative thermocycler. All PCR samples were prepared in triplicate wells in a 96-well plate. After 40 cycles of PCR, melt curves were examined in order to ensure primer specificity. Fold changes in gene expression were calculated using the ΔΔCt method. Primers used for mouse genes were as follows: *Oaz1* (5′-CCACTGCTTCGCCAGAGAG-3′) and (5′-CCCGGACCCAGGTTACTA-3′), *IL-6* (5′-TAGTCCTTCCTACCCCAATTTCC-3′ and 5′-TTGGTCCTTAGCCACTCCTTC-3′), *TNF-α* (5′-CAGGCGGTGCCTATGTCTC-3′ and 5′-CGATCACCCCGAAGTTCAGTAG-3′), *NLRP3* (5′-ATCAACAGGCGAGACCTCTG-3′ and 5′-GTCCTCCTGGCATACCATAGA-3′), *CcL-2* (5′-TTAAAAACCTGGATCGGAACCAA-3′ and 5′-GCATTAGCTTCAGATTTACGGGT-3′), *iNOS* (5′-GTT CTCAGCCCAACAATACAAGA-3′ and 5′-GTGGACGGGTCGATGTCAC-3′), *ABCA1* (5′-TGCCACTTTCCGAATAAAGC-3′ and 5′-GGAGTTGGATAACGGAAGCA-3′), *ABCG1* (5′-AGGTAAAAACCGCCTCCAAG-3′ and 5′-AGTCTGTGTCACCAGGAGCA-3′), *IL-10* (5′-CTTACTGACTGGCATGAGGATCA-3′ and 5′-GCAGCTCTAGGAGCATGTGG-3′), *Arg-1* (5′-CTCCAAGCCAAAGTCCTTAGAG-3′ and 5′-AGG AGCTGTCATTAGGGACATC-3′), CD68 (5′-TGTCTGATCTTGCTAGGACCG-3′ and 5′-GAGAGTAACGGCCTTTTTGTGA-3′), GAPDH (5′-ATGGACTGTGGTCATGAGCC-3′ and 5′-ATTGTCAGCAATGCATCCTG-3′).

Primers used for human genes were as follows: *GAPDH* (5′-CCTCAAGATCATCAGCAATGCCTCCT-3′ and 5′-GGTCATGAGTCCTTCCACGATACCAA-3′), *IL-6* (5′-CCTGAACCTTCCAAAGATGGC-3′) and (5′-TTCACCAGGCAAGTCTCCTCA-3′), *TNF-α* (5′-GAGGCCAAGCCCTGGTATG-3′ and 5′-CGGGCCGATTGATCTCAGC-3′), *FN1* (5′-AGGAAGCCGAGGTTTTAACTG-3′ and 5′-AGGACGCTCATAAGTGTCACC-3′), CD206 (5′-GGGTTGCTATCACTCTCTATGC-3′ and 5′-TTTCTTGTCTGTTGCCGTAGTT-3′).

### 4.4. TLR, NFκB, and AP-1 Activity Assays (Abeomics, Inc., San Diego, CA, USA)

TLR4 Leeporter^TM^ Luciferase Reporter-HeLa cell line: The cell line is a stably transfected HeLa cell line, which expresses full-length human TLR4, MD-2, and CD14 as well as the *Renilla* luciferase reporter gene under the transcriptional control of the IL-8 promoter. The cell line is activated by 6 h treatment with LPS (25 ng/mL). Percent activity was defined as 100 × (1 − (Well − Control A)/(Control B − Control A)), where Control A was the wells treated with LPS and Control B was the wells treated with vehicle alone.

TLR2 Leeporter^TM^ Luciferase Reporter-HEK293 cell line: The cell line is a stably transfected HEK293 cell line, which expresses full-length human TLR2 and *Renilla* luciferase reporter under the transcriptional control of the NFκB response element. The cell line is activated by 6 h treatment with Pam3CSK4 (1 μg/mL). Percent activity was defined as 100 × (1 − (Well − Control A)/(Control B − Control A)), where Control A was the wells treated with Pam3CSK4 and Control B was the wells treated with vehicle alone.

NFκB Leeporter^TM^ Luciferase Reporter-HEK293 cell line: The cell line is a stably transfected HEK293 cell line, which expresses the Renilla luciferase reporter gene under the transcriptional control of the NFκB response element. The cell line is activated by 6 h treatment with PMA (100 ng/mL). Percent activity was defined as 100 × (1 − (Well − Control A)/(Control B − Control A)), where Control A was the wells treated with PMA and Control B was the wells treated with vehicle alone. 

AP-1 LeeporterTM Luciferase Reporter-HEK293 cell line: The cell line is a stably transfected HEK293 cell line, which expresses the Renilla luciferase reporter gene under the transcriptional control of the AP-1 response element. The cell line is activated by 6 h treatment with PMA (100 ng/mL). Percent activity was defined as 100 × (1 − (Well − Control A)/(Control B − Control A)), where Control A was the wells treated with PMA and Control B was the wells treated with vehicle alone. 

### 4.5. COX1 & COX2 Activity Assays (BPS Bioscience, Inc., San Diego, CA, USA)

The test compound was diluted to 10X concentration in 10% DMSO in assay buffer and 10 μL of the dilution was added to a 100 μL reaction so that the final concentration of DMSO from the compounds is 1% in all reactions. All reactions were conducted at room temperature. The reaction mixture contained 60 μL of 1X COX Assay Buffer (BPS Bioscience), 10 μL of enzyme (360 ng COX1 and 250 ng COX2 per reaction) (BPS Bioscience), 10 μL of test compound, 10 μL of 1 mM Ampliflu Red (Sigma-Aldrich), and 10 μL of 0.5 mM arachidonic acid (Cayman Chemical). SC-560 and DuP-697 (Cayman Chemical) were used as reference compounds for inhibiting COX-1 and COX-2, respectively. 

### 4.6. GR and MR Activity Assays (Eurofins DiscoverX, Fremont, CA, USA)

Cell Handling: PathHunter NHR cell lines were expanded from freezer stocks according to standard procedures. Cells were seeded in a total volume of 20 μL into white-walled, 384-well microplates and incubated at 37 °C for the appropriate time prior to testing. Assay media contained charcoal-dextran filtered serum to reduce the level of hormones present. Compound Handling: Sample was diluted into assay buffer shortly before adding to assay. For agonist determination, cells were incubated with sample to induce a response. Intermediate dilution of sample stocks was performed to generate a 5X sample in assay buffer. Five microliters of 5X sample was added to cells and incubated at 37 °C or room temperature for 3–16 h. The final assay vehicle concentration was 1%. For antagonist determination, cells were pre-incubated with antagonist followed by agonist challenge at the EC_80_ concentration. Intermediate dilution of sample stocks was performed to generate a 5X sample in assay buffer. Five microliters of 5X sample was added to cells and incubated at 37 °C or room temperature for 60 min. Vehicle concentration was 1%. Five microliters of 6X EC_80_ agonist in assay buffer was added to the cells and incubated at 37 °C or room temperature for 3–16 h. Signal Detection: Assay signal was generated through a single addition of 12.5 or 15 μL (50% *v*/*v*) of PathHunter detection reagent cocktail, followed by a one-hour incubation at room temperature. Microplates were read following signal generation with a PerkinElmer EnvisionTM instrument for chemiluminescent signal detection. Data Analysis: Compound activity was analyzed using the CBIS data analysis suite (ChemInnovation, CA). For agonist mode assays, percentage activity was calculated using the following formula: % Activity = 100% × (mean RLU of test sample − mean RLU of vehicle control)/(mean MAX control ligand − mean RLU of vehicle control). For antagonist mode assays, percentage inhibition was calculated using the following formula: % Inhibition = 100% × (1 − (mean RLU of test sample − mean RLU of vehicle control)/(mean RLU of EC_80_ control − mean RLU of vehicle control)).

### 4.7. Western Blotting

RAW264.7 cells were plated into 6-well plates at a density of 3 × 10^5^ cells per well. After 2 days of seeding, the cells were pretreated overnight with Oxy210 (5 or 10 μM) or SB431542 (5 μM) in DMEM + 0.1% FBS followed by TGF-β (10 ng/mL) treatment for different times (1 h, 1.5 h or 2 h) or left untreated (0 h). After TGF-β treatment, the cells were harvested and lysed for protein quantitation. Then, 10 μg of protein from each sample was loaded into the wells of an 8% SDS-PAGE gel, and the proteins were transferred to a PVDF membrane after SDS-PAGE gel separation. The membrane was blocked with 5% milk for 1 h at room temperature and then incubated at 4 °C with appropriate dilutions of primary antibodies to pSMAD2 or pSMAD3 (Cell Signaling Technology, Danvers, MA, USA) at 1:1000, or to SMAD2 or SMAD3 (Abcam, Waltham, MA, USA) at 1:1000, or to HSC70 (Santa Cruz Biotechnology, Dallas, TX, USA). After the washing steps, the membrane was incubated with horseradish peroxidase-conjugated secondary antibody at 1:5000 dilution (Cell Signaling Technology) at room temperature for 1 h. West Pico PLUS Chemiluminescent Substrate (Thermo Fisher Scientific, Waltham, MA, USA) was used to develop the signal, and the images were acquired using HyBlot films (Thomas Scientific, Irvine, CA, USA) and a darkroom developer. ImageJ developed by Wayne Rasband at the National Institutes of Health (Bethesda, MD, USA) (version 1.52p) was used for blot densitometry analysis. 

### 4.8. Animal Studies

The breeding and characterization of transgenic mice expressing human cholesteryl ester transfer protein (CETP) and the human APOE*3-Leiden (E3L) were described previously [38]. To generate mice for the fibrosis studies, male C57BL/6J mice carrying both transgenes were bred to BXD19/TyJ females. F1 progeny carrying both transgenes were used for the studies. Animals were maintained on a 12 h light/dark cycle with ad libitum access to food and water. Control mice were fed a “Western” diet (WD, 33 kcal % fat from cocoa butter and 1% cholesterol, Research Diets, cat# D10042101), whereas Oxy210-treated mice were fed the same diet with supplementation of Oxy210 at 4 mg/g or 0.4% (*w*/*w*). Based on an average consumption of 3 g of food by a 30 g mouse per day, this formulation of Oxy210 in food amounts to a dose of 400 mg/kg/day. However, based on the actual average consumption of food during the study, the bona fide dose delivered was about 260 mg/kg/day. Synthesis and characterization of Oxy210 were previously described by MAX BioPharma [36]. Body composition was measured by NMR (Bruker Biospin Corp, Billerica, MA). Euthanasia was carried out with isoflurane vapor followed by cervical dislocation. Gonadal white adipose tissue was surgically excised from all animals, snap-frozen in liquid nitrogen, and stored at −80 °C until RNA extraction when tissues were placed in TRIzol (Thermo Fisher), homogenized, and RNA extracted according to manufacturer’s instructions. All animal work was approved by the UCLA Animal Research Committee, the IACUC.

### 4.9. Statistical Analysis

Statistical analyses were performed using the StatView 5 program (SAS Institute, Cary, NC, USA). All *p* values were calculated using ANOVA and Fisher’s projected least significant difference (PLSD) significance test. A value of *p* < 0.05 was considered significant. The IC_50_ dose-response curves were modeled using a five-parameter logistic model. This model allows for asymmetric curves and automatically estimates the mean maximum and minimum response. Based on this model, IC_50_ values were estimated corresponding to the dose halfway between the minimum and maximum responses. Models of dose versus response and dose versus log response were also evaluated when appropriate. The R square statistic was computed as a measure of model fit. Differences between two groups in the animal studies were compared using the Mann–Whitney *t*-test. *p* values < 0.05 were considered significant.

## 5. Conclusions

Oxy210 is a semi-synthetic oxysterol with anti-inflammatory properties in white adipose tissue and liver in vivo, and in macrophages in vitro, mediated in part through inhibition of TLR4, TLR2, and AP-1 signaling. Through its TLR-modulating effects, Oxy210 and some of its analogs may serve as drug candidates for intervention in chronic inflammatory diseases that are mediated by activation of the innate immune system. Oxy210 presents the unique property of inhibiting both M1 and M2 macrophage polarization which may be therapeutically helpful in conditions driven by chronic inflammation and fibrosis. These in vivo and in vitro data, along with encouraging oral bioavailability, tolerability, and disease-modifying properties observed in vivo, suggest that Oxy210 may be a favorable drug candidate for further therapeutic development.

## Figures and Tables

**Figure 1 ijms-23-05478-f001:**
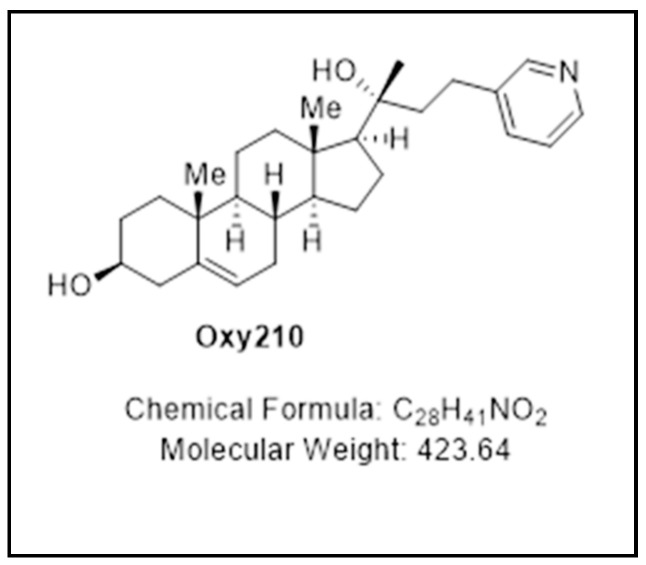
Molecular structure, chemical formula, and molecular weight of Oxy210.

**Figure 2 ijms-23-05478-f002:**
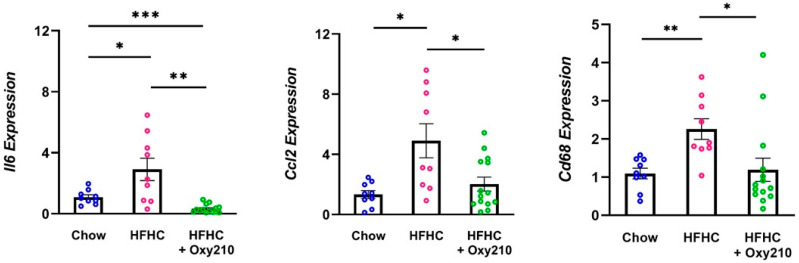
Effects of Oxy210 on pro-inflammatory gene expression in gonadal white adipose tissue. Expression of pro-inflammatory genes, *Il6* and *Ccl2*, and macrophage marker, *Cd68*, in white adipose tissue from control (chow, *n* = 9), high fat high cholesterol (HFHC, *n* = 9) and HFHC + Oxy210-fed (*n* = 14) mice were measured by Q-RT-PCR and normalized to GAPDH. Relative gene expression levels are presented as the mean ± SD. (* *p* < 0.05; ** *p* < 0.01; *** *p* < 0.001).

**Figure 3 ijms-23-05478-f003:**
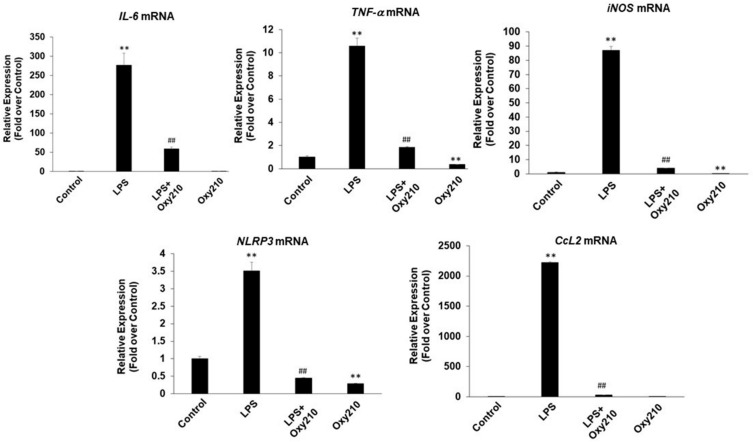
Inhibition of LPS-induced inflammatory gene expression in RAW264.7 macrophages by Oxy210. RAW264.7 cells were pretreated with Oxy210 (5 μM) in DMEM containing 0.1% FBS for 24 h followed by the addition of LPS (25 ng/mL) in the absence or presence of Oxy210 (5 μM). The 5 μM concentration of Oxy210 was optimized for causing maximum inhibitory effects on all outcomes measured. After 24 h, RNA was extracted and analyzed by Q-RT-PCR for the expression of inflammatory genes as indicated and normalized to *Oaz1* expression. Data from a representative experiment are reported as the mean of triplicate determinations ± SD (** *p* < 0.01 vs. control; ## *p* < 0.01 vs. LPS).

**Figure 4 ijms-23-05478-f004:**
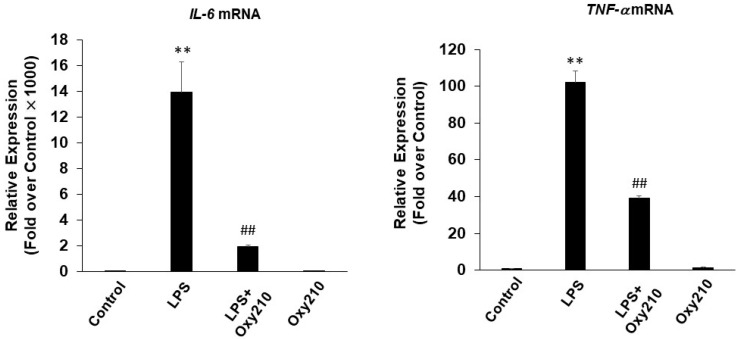
Inhibition of LPS-induced inflammatory gene expression in THP-1 macrophages by Oxy210. THP-1 cells were pretreated with Oxy210 (5 μM) in RPMI containing 0.1% FBS for 24 h followed by the addition of LPS (100 ng/mL) in the absence or presence of Oxy210 (5 μM). After 24 h, RNA was extracted and analyzed by Q-RT-PCR for the expression of inflammatory genes as indicated and normalized to GAPDH expression. Data from a representative experiment are reported as the mean of triplicate determinations ± SD (** *p* < 0.01 vs. control; ## *p* < 0.01 vs. LPS).

**Figure 5 ijms-23-05478-f005:**
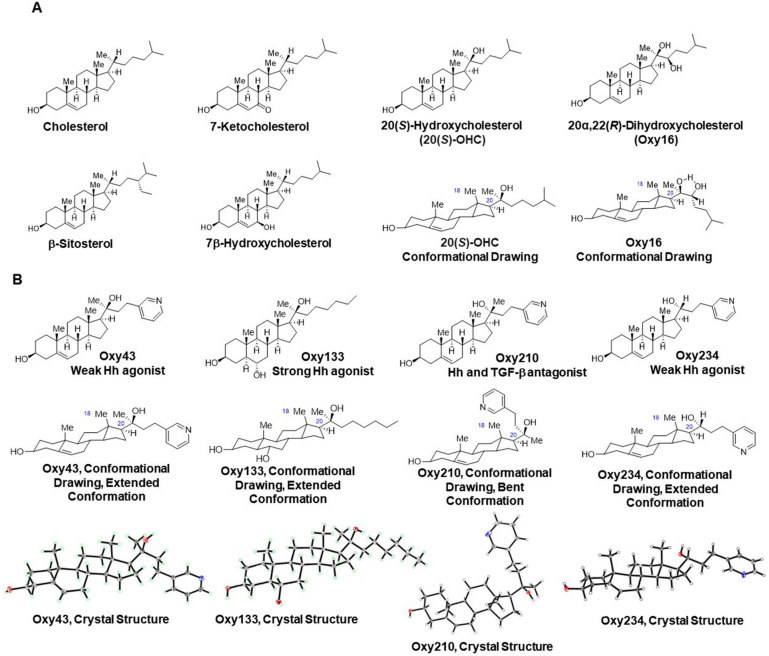
(**A**) Molecular structures of naturally occurring sterols and oxysterols. Cholesterol, β-sitosterol, 7-ketocholesterol, 7β-hydroxycholesterol, 20(*S*)-hydroxycholesterol (20(*S*)-OHC), and 20α,22(*R*)-dihydroxycholesterol (Oxy16). Conformational drawings of 20(*S*)-OHC and Oxy16 are based on crystal structures. (**B**) Molecular structures, conformational drawings, and crystal structures of semi-synthetic oxysterols Oxy43, Oxy133, Oxy210, and Oxy234.

**Figure 6 ijms-23-05478-f006:**
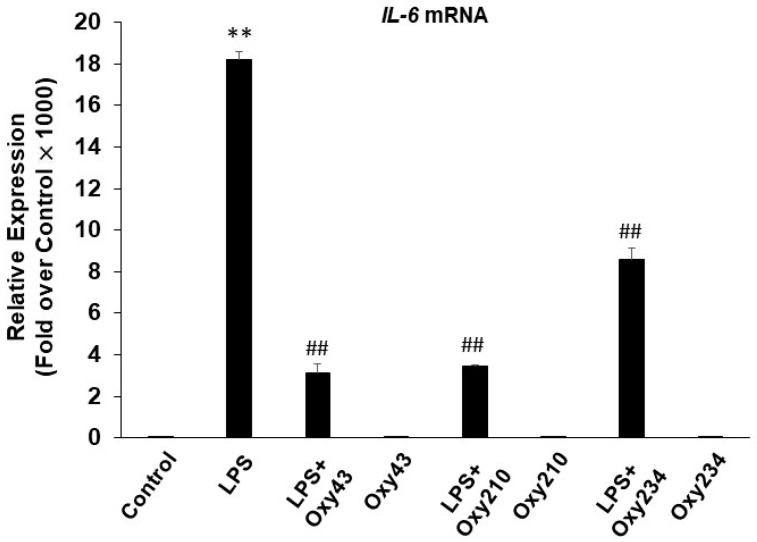
Inhibition of LPS-induced inflammatory gene expression in RAW264.7 macrophages by oxysterols. RAW264.7 cells were pretreated with the oxysterols (5 μM) as indicated in DMEM containing 0.1% FBS for 24 h followed by the addition of LPS (25 ng/mL) in the absence or presence of the oxysterols (5 μM). After 24 h, RNA was extracted and analyzed by Q-RT-PCR for the expression of IL-6 and normalized to *Oaz1* expression. Data from a representative experiment are reported as the mean of triplicate determinations ± SD (** *p* < 0.01 vs. control; ## *p* < 0.01 vs. LPS).

**Figure 7 ijms-23-05478-f007:**
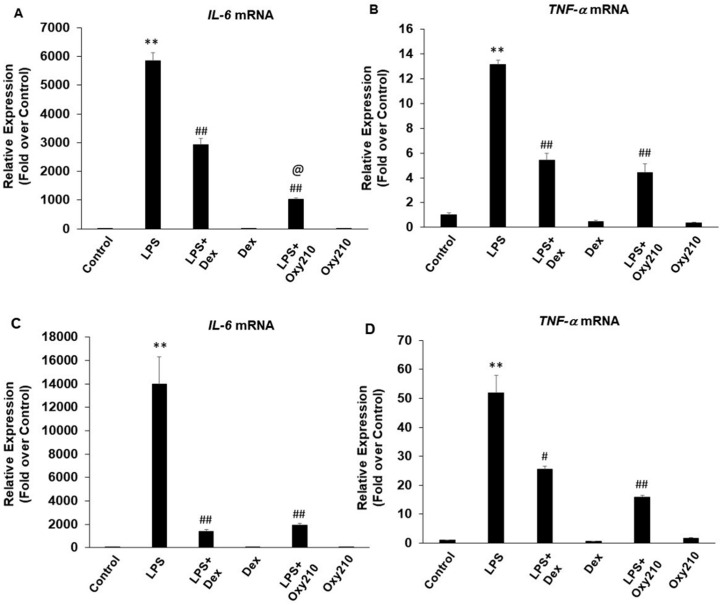
Inhibition of LPS-induced inflammatory gene expression in RAW264.7 macrophages by dexamethasone and Oxy210. (**A**,**B**) RAW264.7 cells were pretreated with Oxy210 (5 μM) or dexamethasone (Dex) (5 μM) in DMEM containing 0.1% FBS for 24 h followed by the addition of LPS (25 ng/mL) in the absence or presence of Oxy210 (5 μM) or Dex (5 μM). (**C**,**D**) THP-1 cells were pretreated with Oxy210 (5 μM) or Dex (5 μM) in RPMI containing 0.1% FBS for 24 h followed by the addition of LPS (100 ng/mL) in the absence or presence of Oxy210 or Dex (5 μM). After 24 h, RNA was extracted and analyzed by Q-RT-PCR for the expression of the genes as indicated and normalized to *Oaz1* (**A**,**B**) or GAPDH (**C**,**D**) expression. Data from a representative experiment are reported as the mean of triplicate determinations ± SD (** *p* < 0.01 vs. control; # *p* < 0.05 vs. LPS; ## *p* < 0.01 vs. LPS; @ *p* < 0.001 vs. LPS+Dex).

**Figure 8 ijms-23-05478-f008:**
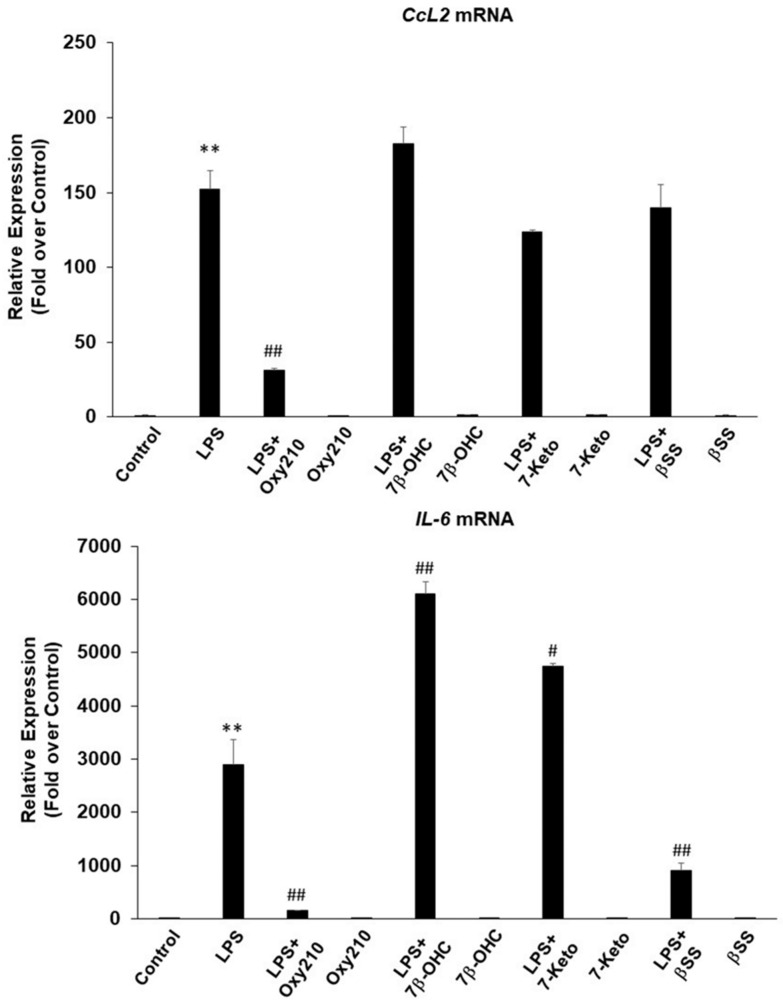
Comparison of Oxy210 with naturally occurring sterols and oxysterols for their inhibitory effects on LPS-induced inflammatory gene expression in RAW264.7 macrophages. RAW264.7 cells were pretreated with the compounds (5 μM) as indicated in DMEM containing 0.1% FBS for 24 h followed by the addition of LPS (25 ng/mL) in the absence or presence of the compounds (5 μM). After 24 h, RNA was extracted and analyzed by Q-RT-PCR for the expression of the genes as indicated and normalized to *Oaz1* expression. Data from a representative experiment are reported as the mean of triplicate determinations ± SD (** *p* < 0.01 vs. control; # *p* < 0.05 vs. LPS; ## *p* < 0.01 vs. LPS).

**Figure 9 ijms-23-05478-f009:**
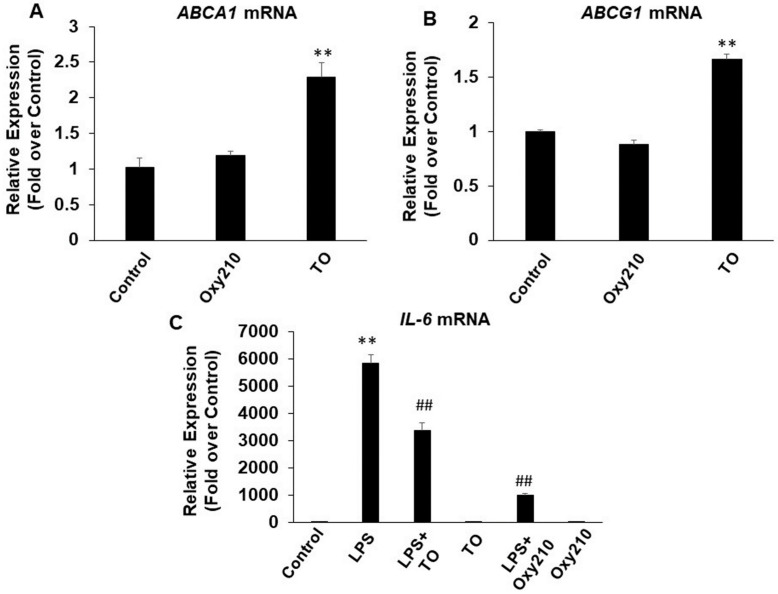
Induction of LXR target gene expression and anti-inflammatory effects of Oxy210 and LXR agonist TO901317 (TO) in RAW264.7 macrophages. (**A**,**B**) RAW264.7 cells were treated with Oxy210 or TO (each at 5 μM) in DMEM containing 0.1% FBS for 48 h. (**C**) RAW264.7 cells were pretreated with Oxy210 or TO (each at 5 μM) as indicated in DMEM containing 0.1% FBS for 24 h followed by the addition of LPS (25 ng/mL) in the absence or presence of the compounds (5 μM) for 24 h. RNA was extracted and analyzed by Q-RT-PCR for the expression of the genes as indicated and normalized to *Oaz1* expression. Data from a representative experiment are reported as the mean of triplicate determinations ± SD (** *p* < 0.01 vs. control; ## *p* < 0.01 vs. LPS).

**Figure 10 ijms-23-05478-f010:**
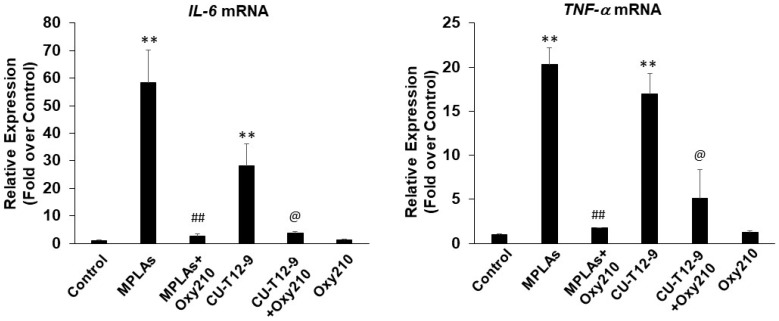
Inhibition of TLR4 and TLR2 agonist-induced inflammatory gene expression by Oxy210 in RAW264.7 macrophages. RAW264.7 cells were pretreated with Oxy210 (5 μM) in DMEM containing 0.1% FBS for 24 h followed by the addition of TLR4 agonist, monophosphoryl lipid A (MPLAs) (1 μg/mL), or TLR2 agonist, CU-T12-9 (1 μg/mL), in the absence or presence of Oxy210 (5 μM). After 24 h, RNA was extracted and analyzed by Q-RT-PCR for the expression of the genes as indicated and normalized to *Oaz1* expression. Data from a representative experiment are reported as the mean of triplicate determinations ± SD (** *p* < 0.01 vs. control; ## *p* < 0.01 vs. MPLAs; @ *p* < 0.05 vs. CU-T12-9).

**Figure 11 ijms-23-05478-f011:**
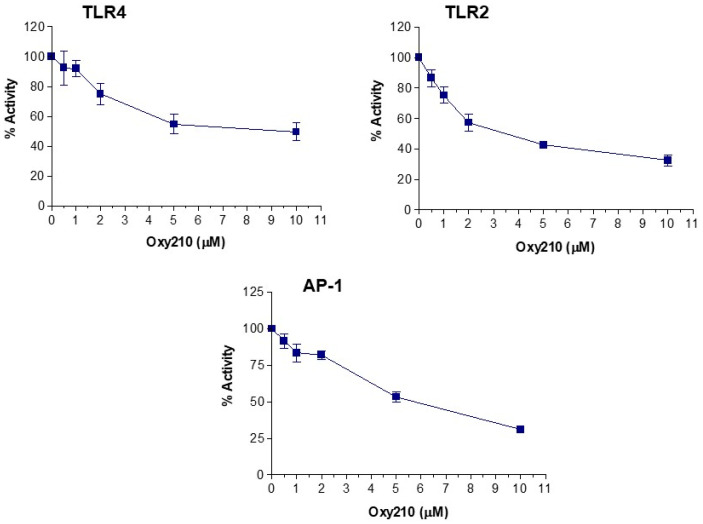
Inhibition of TLR4, TLR2, and AP-1 driven luciferase reporter activity by Oxy210. TLR4 Leeporter^TM^ Luciferase Reporter-HeLa cells, TLR2 Leeporter^TM^ Luciferase Reporter-HEK293 cells, and AP-1 Leeporter^TM^ Luciferase Reporter-HEK293 cells were treated with TLR4 agonist, LPS (25 ng/mL), TLR2 agonist, Pam3CSK4 (1 μg/mL), or AP-1 activator, PMA (100 ng/mL), respectively, for 6 h in the presence or absence of Oxy210 at the concentrations indicated. Luciferase activity was measured and normalized to the Renilla luciferase activity. Data reported as % activity calculated from triplicate wells per Oxy210 dose ± SD as described in Materials and Methods.

**Figure 12 ijms-23-05478-f012:**
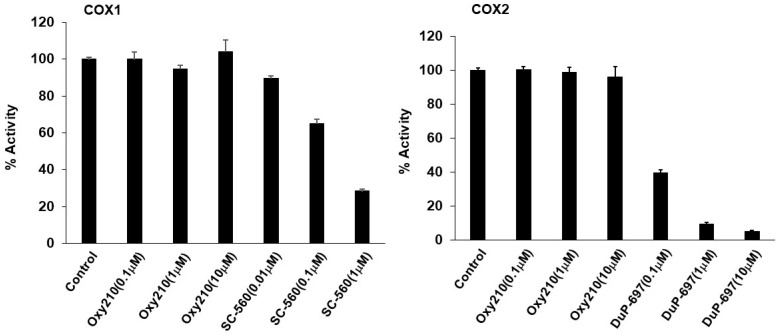
Effects of Oxy210 on COX1 and COX2 enzyme activities. The reactions were performed as described in Materials and Methods using the compounds as indicated. Data are reported as the mean of triplicate determinations ± SD.

**Figure 13 ijms-23-05478-f013:**
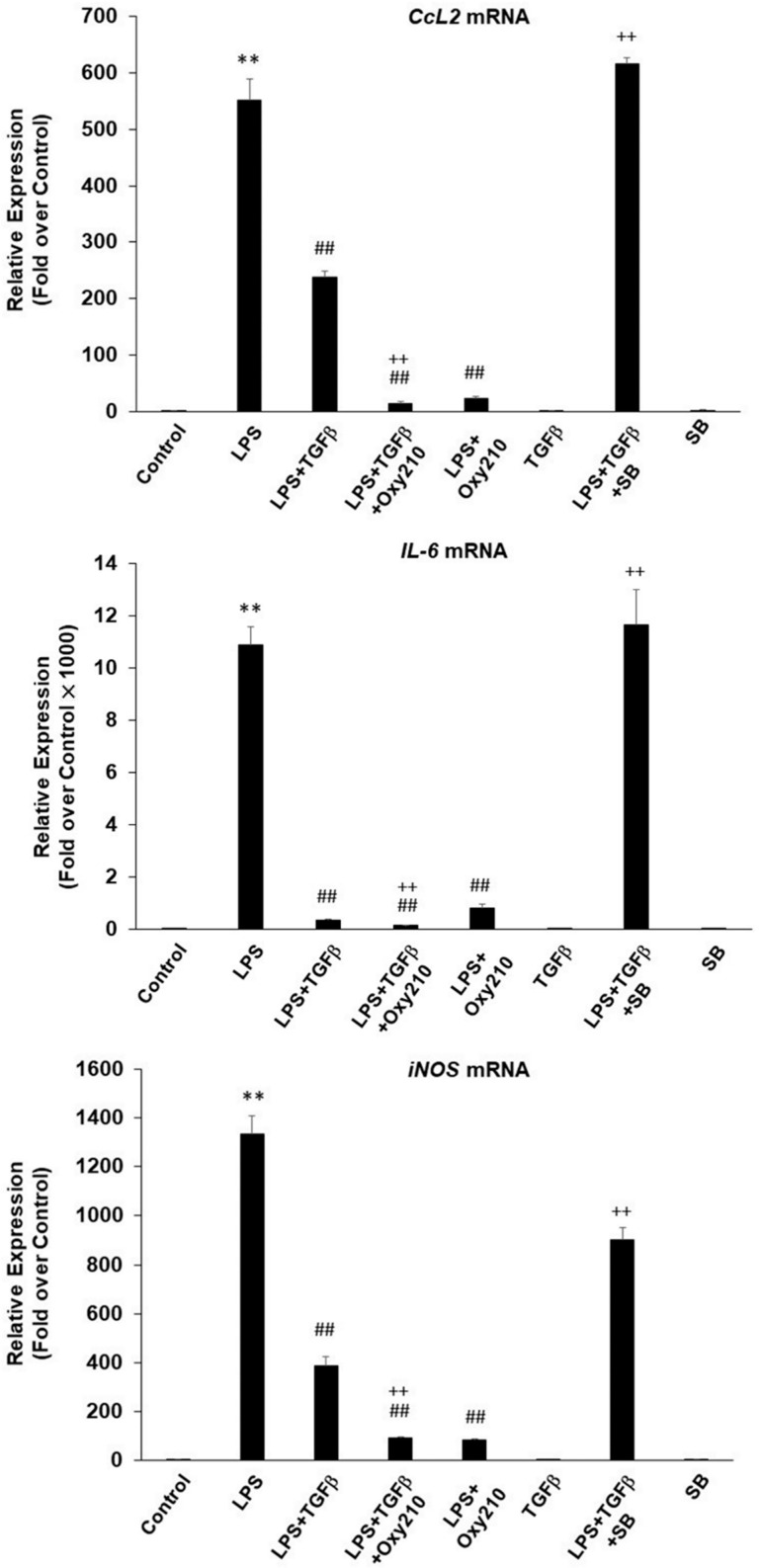
Enhancement of the anti-inflammatory effect of TGF-β in RAW264.7 mouse macrophages by Oxy210. RAW264.7 cells were pretreated with Oxy210 (5 μM) or SB431542 (SB) (5 μM) in DMEM containing 0.1% FBS for 24 h followed by the addition of TGF-β (10 ng/mL) for 2 h after which LPS (25 ng/mL) was added. After 24 h, RNA was extracted and analyzed by Q-RT-PCR for the expression of the genes as indicated and normalized to *Oaz1*. Data from a representative experiment are reported as the mean of triplicate determinations ± SD (** *p* < 0.01 vs. control; ## *p* < 0.01 vs. LPS; ++ *p* < 0.01 vs. LPS+TGF-β).

**Figure 14 ijms-23-05478-f014:**
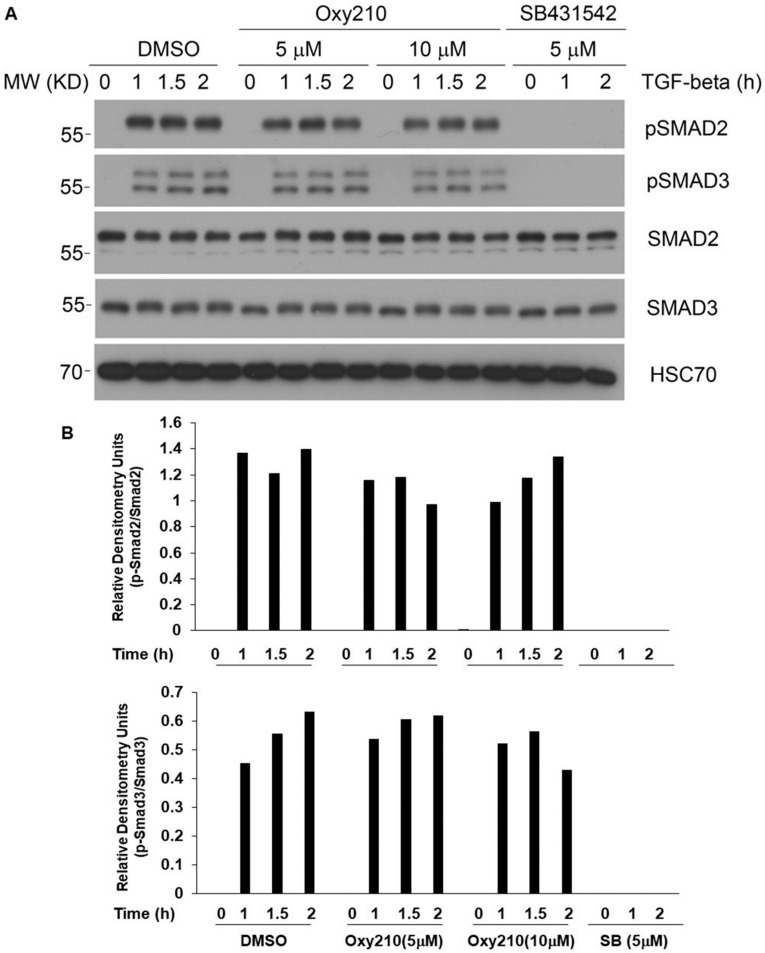
Effects of Oxy210 on TGF-β induced Smad2/3 phosphorylation. (**A**) RAW264.7 cells were pretreated with DMSO, Oxy186, Oxy210, or SB431542 overnight, and then treated with TGF-β (10 ng/mL) for the indicated times in the presence of the compounds as indicated. (**B**) Phosphorylated Smad2/3 were quantified using ImageJ and normalized to their corresponding non-phosphorylated version. Data from a representative of two separate experiments are shown.

**Figure 15 ijms-23-05478-f015:**
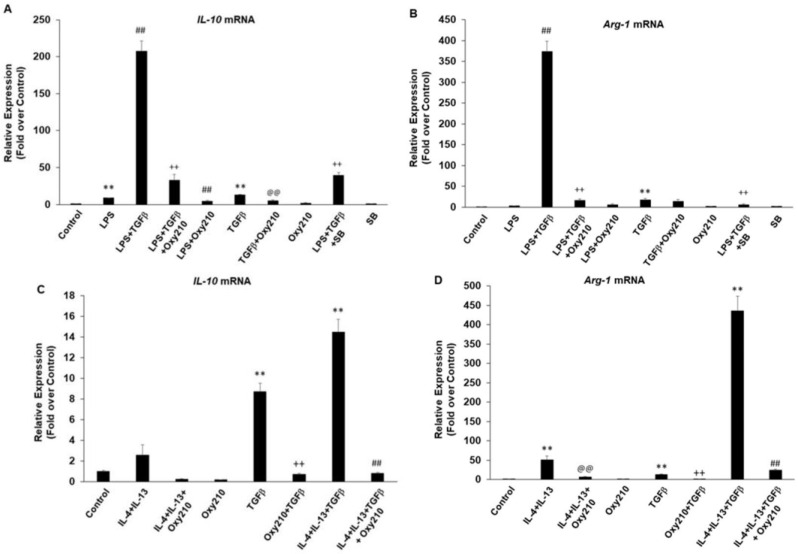
Inhibition of LPS, TGF-β, and IL-4+IL-13 induced M2 polarization of RAW264.7 macrophages by Oxy210. Cells were pretreated with Oxy210 (5 μM) or SB431542 (5 μM) in DMEM containing 0.1% FBS for 24 h followed by the addition of LPS (25 ng/mL) (Figure 13A,B), TGF-β (10 ng/mL) (**A**–**D**), or IL-4+IL-13 (10 ng/mL each) (**C**,**D**). After 24 h, RNA was extracted and analyzed by Q-RT-PCR for the expression of the genes as indicated and normalized to *Oaz1*. Data from a representative experiment are reported as the mean of triplicate determinations ± SD (** *p* < 0.01 vs. control; ## *p* < 0.01 vs. IL-4+IL-13+TGF-β; @@ *p* < 0.01 vs. IL-4+IL-13; ++ *p* < 0.01 vs. TGF-β).

**Figure 16 ijms-23-05478-f016:**
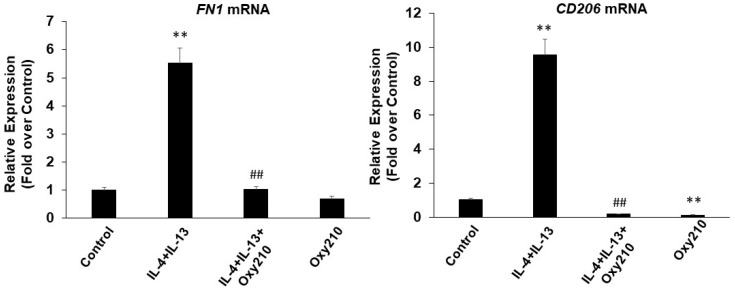
Inhibition of IL-4+IL-13 induced M2 polarization of THP-1 macrophages by Oxy210. Cells were pretreated with Oxy210 (2.5 μM) in RPMI containing 0.1% FBS for 24 h followed by the addition of IL-4+IL-13 (20 ng/mL each) in the absence or presence of Oxy210 (2.5 μM). After 48 h, RNA was extracted and analyzed by Q-RT-PCR for the expression of the genes as indicated and normalized to GAPDH. Data from a representative experiment are reported as the mean of triplicate determinations ± SD (** *p* < 0.01 vs. control; ## *p* < 0.01 vs. IL-4+IL-13).

**Figure 17 ijms-23-05478-f017:**
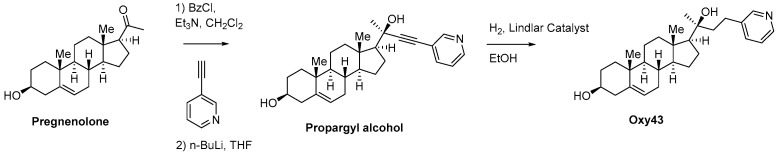
Synthesis of Oxy43.

**Figure 18 ijms-23-05478-f018:**
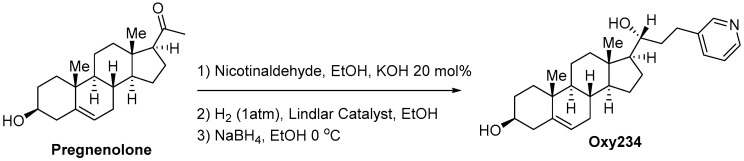
Synthesis of Oxy234.

**Table 1 ijms-23-05478-t001:** Inhibition of inflammatory cytokine expression by Oxy210 in RAW264.7 macrophages. Cells were pretreated for 24 h with serial dilutions of Oxy210 (0.5, 1, 2.5, 5, 10 μM) in DMEM containing 0.1% FBS followed by the addition of LPS (25 ng/mL) in the absence or presence of Oxy210 at the same concentrations as that used in pretreatment. After 24 h, RNA was extracted and analyzed by Q-RT-PCR for the expression of the inflammatory genes as indicated and normalized to Oaz1 expression. IC_50_ values for the inhibition of each gene from a representative experiment are reported as the mean of triplicate determinations ± SD.

Gene	IC_50_ (µM)
*IL-6*	0.99 ± 0.36
*TNF-a*	1.67 ± 0.13
iNOS	1.15 ± 0.58
MCP-1 (CCL-2)	1.07 ± 0.07
NLRP3	1.73 ± 0.33

**Table 2 ijms-23-05478-t002:** Effect of Oxy210 on glucocorticoid (GR) and mineralocorticoid receptor (MR) activity. The GR and MR activity assays were performed as described in Materials and Methods using the compounds as indicated. Dexamethasone and aldosterone were used as positive controls for GR and MR, respectively.

	GR-Agonism	MR-Agonism
Dexamethasone (10 µM)	100%	-
Dexamethason (0.4 µM)	100%	-
Aldosteron (1 µM)	-	100%
Aldosteron (0.04 µM)	-	100%
Oxy210 (10 µM)	1%	−3.6%

## Data Availability

Not applicable.

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
