# Peer review of "Oxy210, a Semi-Synthetic Oxysterol, Exerts Anti-Inflammatory Effects in Macrophages via Inhibition of Toll-like Receptor (TLR) 4 and TLR2 Signaling and Modulation of Macrophage Polarization"

_ijms, 2022, doi:10.3390/ijms23105478_

Round 1

Reviewer 1 Report

The submitted manuscript describes a wide range of assays concerning the anti-inflammatory activity of semi-synthetic oxysterol such as Oxy210. The following of the manuscript by readers might be difficult. The Authors tested the gene expression of inflammatory mediators in white adipose tissue of APOE*3-Leident. CETP mice and cell lines of murine and human macrophages (RAW264.7 and THP-1). The same factors were studied in both cell lines, e.g. IL-6 and TNFα mRNAs. In some experiments, Oxy43 and Oxy234 were included in the study but it seems that these results did not determine any further experiments. In Figure 4 (L. 257 Figure 3?) the structures of different sterols were displayed although the study was focused only on Oxy210. Figures 4 and 5 are completely unnecessary. The Authors attempted to establish the relation of structure to activity (L. 257-265) but it requires more experiments. Thus, the manuscript is very confusing. Firstly,  in many experiments Oxy210 was tested only in one concentration. Therefore, there is no conclusion if the effect depends on concentration. In addition, the same data are often provided in two figures or at least it is not clear (Figure 2 and 6). Macrophages of M2 phenotype are characterized by the anti-inflammatory activity, e.g. by secretion of IL-10. Oxy210 inhibited IL-10 mRNA, which should not be strictly treated as positive effect. It was cited that β-sitosterol promotes M2 anti-inflammatory polarization. This could not have been observed in the case of Oxy210.

The section on Materials and methods does not provide sufficient data on experiments which does not allow for assessment of them. Some sections are not necessary (4.2.1, 4.2.2., 4.2.4). Sections 4.4, 4.6, 4.7 are confusing.

Author Response

Dear Editors and Reviewers:

Thanks for reviewing our manuscript “Oxy210, a semi-synthetic oxysterol, exerts anti-inflammatory effects in macrophages via inhibition of Toll-Like Receptor (TLR) 4 and TLR2 signaling and modulation of macrophage polarization” for publication in the International Journal of Molecular Sciences.  We are grateful for the comments made by the reviewers and we have revised the manuscript to address all comments and suggestions, and as a result, we believe that the manuscript has been significantly improved.  The following are point-by-point responses to the reviewers’ comments and explanation of all the revisions that we have made, which are also seen in the manuscript text using the “Track Changes” function as you have instructed. 

Reviewer 2 Report

This manuscript explores the potential mechanisms of anti-inflammatory properties of the steroid-pyridine-modified named oxy210. The compound mimics the anti-inflammatory effects of dexamethasone in RAW264.7 cultured cells stimulated with LPS, via LXR activation, and by inhibiting AP-1 TLR-1/2 and TLR-4 signaling, but not via NF-kB signaling or Hhg pathway. Moreover, oxy210 inhibited both, M1 and M2 macrophage polarization by a mechanism that involve TGF-beta receptor but independently of SMAD 1/2 phosphorylation.

The manuscript sounds scientifically valid, most of the potential pathways involved in the mechanisms of action of oxy210 were explored. However, the manuscript is difficult to follow, and there are some issues that should be addressed:

  1. The introduction section is too long. Condense the information to the just the necessary to justify the aim of the study and limit the results in this section [lines 106 to 134] to only 2 or 3 lines.
  2. Section 2.1. There is no detail about neither the adipose tissue used in this study nor the method of handling and extraction of tissular RNA. Was the sample obtained from subcutaneous, visceral, epididymal or other adipose white tissue? The distinct adipose tissues are metabolically and endocrinology different among them, and inflammation is particularly important in visceral one.
  3. Figure 1. Why different number of animals per group (8, 9 and 14)? A non-parametric test to compare the groups is advisable under such conditions.
  4. Did authors really used 0.1% of fetal bovine serum in the culture media? The cells used in this study does not grow at that FBS concentrations.
  5. Section 2.2/Table 1- It is very difficult to understand why authors reported RNAm expression as IC50. It is in fact incorrect. Table 1 indicates the concentration at which the gene expressions decreased at 50% of the control, but this information is not clear. Include, i.e as supplementary material, the dose-response curves of the different concentrations of oxy210 used for these experiments and change the foot table to provide accurate information.
  6. Lines 173 to 199 should be condensed; these details may be included in discussion section, but in results section the justification of the next experiments should be focused with only a couple of lines.
  7. Molecular structures, of Oxy43, Oxy133, Oxy210 and Oxy234 may be presented in the first subsection of the results to make the text more readable.
  8. Line 173, the therapeutic concentrations of ibuprofen are not the same than those of dexamethasone; ibuprofen pharmacological concentrations much more elevated than 5 μM. These experiments could not be considered as valid because of the very low concentrations used.
  9. Lines 198-304. The information provided:

Oxy210 298 did significantly induce ABCA1 and ABCG1 expression in human THP-1 macrophages 299 by 9 fold and 53 fold, respectively, after 48 hours under similar experimental conditions to those used with murine RAW264.7 cells (data not shown). Given the absence of any correlation between LXR activation and anti-inflammatory effects in RAW264.7 and THP-1 macrophages, these findings suggest that the anti-inflammatory effects of Oxy210 in 303 macrophages are likely through LXR-independent mechanisms.”

is contradictory; Authors first indicated that oxy210 highly increases ABCA1 and ABCG1 expressions (LXR dependent genes) and a few lines below they stated that the anti-inflammatory effects of Oxy210 in  macrophages are likely through LXR-independent mechanisms. Correct this incongruency and discuss the possibility that oxy210 anti-inflammatory effects may also include LXR activation in human cells.

  1. As indicated in figure legends, most results presented in graphs are the data of one representative experiment in triplicate. Are the statistical analyses performed considering only those triplicates of the same experiment? If so, the results are not strong enough and represent a weakness of the study that should be recognized in the manuscript.
  2. Lines 271-273. The statement “Results showed that at 5 mM, Oxy210 was consistently more effective than 5 mM dexamethasone in inhibiting LPS-induced IL-6 gene expression and as effective in inhibiting TNF-a expression in RAW264.7 cells (Figure 6A,B),” is not congruent with figure 6; in figure 6, the level of IL6 and TNF mRNA expression is similar with dexamethasone and with Oxy210, at least there is not any statistical difference between these two compounds indicated in such figure; oxy210 is not more effective that dexamethasone as stated by the Authors. Correct this incongruency and try to interpret data with this result in mind.
  3. Figure 10, 1st graph. TLR4 driven luciferase reporter activity in the presence of Oxy210 does not seem to reach the 50% of inhibition (that should be around 45 fold change) even with the highest dose presented in this graph; it is not clear how Authors could calculate an IC50 of 2.2 μM. Explain this issue and if necessary reinterpret data the lack of inhibition of TLR4 signaling in these experiments.
  4. Lines 410-412. The statement “Treatment of RAW cells with TGF-b inhibited LPS-induced IL-6, CCL2 and iNOS expression, an effect that was further enhanced, rather than inhibited, in the presence of Oxy210 (Figure 12)” is incorrect, Figure 12 clearly shows that Oxy210 does not enhance gene expressions, they remained the same. Correct this statement and include this results in the overall data interpretation.
  5. Figure 13. How many experiments were included for this figure? Just one without triplicate? There is not any dispersion bar in this figure.
  6. Does oxy210 act extracellularly as an antagonist of TLR1/2 an TLR4 or intracellularly by for example interacting with transcription cofactors or both? Could authors speculate a little bit about this issue?

Author Response

(The authors gave the same response as above.)

Round 2

Reviewer 2 Report

My main concerns were satisfactorily addressed. 

Author Response

Dear Editors and Reviewers:
Thanks for reviewing our revised manuscript “Oxy210, a semi-synthetic oxysterol, exerts anti-inflammatory effects in macrophages via inhibition of Toll-Like Receptor (TLR) 4 and TLR2 signaling and modulation of macrophage polarization” for publication in the International Journal of Molecular Sciences. We are very happy that in our revised manuscript we had successfully addressed all the main concerns of the reviewers.
Please accept a second revision of our manuscript in which we have addressed all the minor comments made by the reviewers:
1. We have done a thorough review and have corrected any spelling errors and typos as suggested by the reviewer.
2. We have checked and made sure that all references are relevant to the research being presented.
3. The previously shortened Introduction section (as previously suggested by the reviewers) has all the necessary background information for the research being presented.
Please note: In the second revision, Figures 7 and 11 still need to be replaced as we had previously requested in the first revision. The revised Figures addressed the previous comments of the reviewers.
We look forward to the acceptance of our manuscript for publication in your distinguished Journal. Please let us know if any other information is necessary.